# Does Plant Breeding for Antioxidant-Rich Foods Have an Impact on Human Health?

**DOI:** 10.3390/antiox11040794

**Published:** 2022-04-18

**Authors:** Laura Bassolino, Katia Petroni, Angela Polito, Alessandra Marinelli, Elena Azzini, Marika Ferrari, Donatella B. M. Ficco, Elisabetta Mazzucotelli, Alessandro Tondelli, Agostino Fricano, Roberta Paris, Inmaculada García-Robles, Carolina Rausell, María Dolores Real, Carlo Massimo Pozzi, Giuseppe Mandolino, Ephrem Habyarimana, Luigi Cattivelli

**Affiliations:** 1CREA—Research Centre for Cereal and Industrial Crops, Via di Corticella 133, 40128 Bologna, Italy; laura.bassolino@crea.gov.it (L.B.); roberta.paris@crea.gov.it (R.P.); giuseppe.mandolino@crea.gov.it (G.M.); e.habyarimana@cgiar.org (E.H.); 2Department of Biosciences, University of Milan, Via Celoria 26, 20133 Milan, Italy; katia.petroni@unimi.it (K.P.); alessandra.marinelli@guest.unimi.it (A.M.); 3CREA—Research Centre for Food and Nutrition, Via Ardeatina 546, 00178 Rome, Italy; angela.polito@crea.gov.it (A.P.); elena.azzini@crea.gov.it (E.A.); marika.ferrari@crea.gov.it (M.F.); 4CREA—Research Centre Cereal and Industrial Crops, S.S. 673 Km 25,200, 71122 Foggia, Italy; donatellabm.ficco@crea.gov.it; 5CREA—Research Centre for Genomics and Bioinformatics, Via San Protaso 302, 29017 Fiorenzuola d’Arda, Italy; elisabetta.mazzucotelli@crea.gov.it (E.M.); alessandro.tondelli@crea.gov.it (A.T.); agostino.fricano@crea.gov.it (A.F.); 6Department of Genetics, University of Valencia, Dr. Moliner 50, 46100 Burjassot, Spain; inmaculada.garcia@uv.es (I.G.-R.); carolina.rausell@uv.es (C.R.); maria.dolores.real@uv.es (M.D.R.); 7Department of Agricultural and Environmental Sciences-Production, Landscape, Agroenergy, University of Milan, Via Celoria 26, 20133 Milan, Italy; carlo.pozzi@unimi.it; 8International Crops Research Institute for the Semi-Arid Tropics, Patancheru 502324, India

**Keywords:** antioxidants, polyphenols, carotenoids, cereals, *Solanaceae*, pre-clinical studies, food diet

## Abstract

Given the general beneficial effects of antioxidants-rich foods on human health and disease prevention, there is a continuous interest in plant secondary metabolites conferring attractive colors to fruits and grains and responsible, together with others, for nutraceutical properties. Cereals and *Solanaceae* are important components of the human diet, thus, they are the main targets for functional food development by exploitation of genetic resources and metabolic engineering. In this review, we focus on the impact of antioxidants-rich cereal and *Solanaceae* derived foods on human health by analyzing natural biodiversity and biotechnological strategies aiming at increasing the antioxidant level of grains and fruits, the impact of agronomic practices and food processing on antioxidant properties combined with a focus on the current state of pre-clinical and clinical studies. Despite the strong evidence in in vitro and animal studies supporting the beneficial effects of antioxidants-rich diets in preventing diseases, clinical studies are still not sufficient to prove the impact of antioxidant rich cereal and *Solanaceae* derived foods on human

## 1. Introduction

The tremendous variety of fruits and grains that provide colour to cereals and *Solanaceae*, respectively, is mainly due to carotenoids and polyphenols (e.g., flavonoids), two classes of phytochemicals characterized by antioxidant properties. These bioactive compounds, together with ascorbic acid, vitamin E, and others, are considered health-promoting components of the human diet, given their capacity of exerting scavenging activity on reactive oxygen species (ROS) and free radicals, protecting cells and tissues from oxidative stress, and the onset of chronic diseases. Polyphenols exert a well-known beneficial effect on gut microbiota and, at cellular level, trigger the signaling cascade of endogenous antioxidant defences [1,2]. In many crops, the great diversity in fruit and grain colour allows the selection of cultivars with different amounts and types of pigments. Plant breeding has invested significant resources to select pigmented varieties with a higher content in carotenoids and flavonoids and, consequently, higher antioxidant potential [3]. Cereals and *Solanaceae* are relevant dietary sources of nutrients in the human diet [4] with a vast and well characterized genetic diversity. The available genetic knowledge could allow replacing the current varieties with new ones enriched in pigments and the antioxidant content of food crops could even be reinforced by using specific crop managements. These efforts could translate into vegetables and grains with an increased nutritional value that may result in antioxidant-rich foods.

This review aims at exploring genetics and new cereals and *Solanaceae* breeding programs for antioxidant-rich foods, particularly carotenoids and polyphenols, together with the effect of agronomic practices and food processing on levels of these compounds, and their potential impact on human health.

## 2. The Pigment Biosynthetic Pathways

### 2.1. The Conserved Flavonoid Pathway

Flavonoids share a general C3–C6–C3 structure with two aromatic rings linked by a three-carbon bridge, where the degree of oxidation of the C-ring defines the various flavonoid sub-classes. They are synthesized through the phenylpropanoid pathway [5,6] (Figure 1), consisting of several branches that, starting from the aromatic amino acid phenylalanine, lead to the synthesis of distinct classes of compounds: anthocyanins, proanthocyanidins (PA or condensed tannins), phlobaphene pigments and the non-pigmented (i.e., not visible to human eye) flavonols, flavones, and isoflavones [7]. Both flavones and flavonols generally have little colour, but they may influence anthocyanin colour through co-pigmentation. The number of hydroxyl groups on the B-ring (side chain decorations) of the flavonoids influences their ability to scavenge different free radicals [8], whereas the stability of anthocyanins is highly influenced by glycosylation and acylation. As an example, anthocyanin 3-disaccharides are generally more stable than 3-monosaccharides and 3,5-disaccharides, and acylation of sugars with aromatic or aliphatic acids further increases their stability, allowing co-pigmentation and a shift towards blue [9]. In general, ferulic acid is predominantly present in cereal grains, whereas chlorogenates are the major phenolic compounds found in solanaceous fruits. In cereals, pigmented tissues mainly accumulate cyanidin and to a lesser extent pelargonidin-derived anthocyanins or phlobaphenes in corn pericarp (Figure 1a). Delphinidin and pelargonidin in the form of anthocyanidin-3-(p-coumaroyl-rutinoside)-5-glucoside are the predominant type in *Solanaceae*, whereas the latter is restricted to red potato tubers [10] (Figure 1b). The flavonoid pathway is largely regulated at transcriptional level by transcription factors which modulate the spatio-temporal expression of biosynthetic genes [11]. R2R3-MYBs are considered the master regulators of the flavonoid pathway, since they can act as independent regulatory proteins, thereby activating biosynthetic genes and other regulators, e.g., bHLHs. Furthermore, they can also act as members of multimeric complexes known as MBW, consisting of R2R3-MYBs, bHLHs, and WD40s [12]. 

A different regulation of flavonoid biosynthesis occurs in monocots and dicots. In monocots, all anthocyanin biosynthetic genes are coordinately activated by MYB and bHLH proteins In Figure 1a, the biosynthetic pathway leading to the diverse classes of polyphenols in monocots and the major regulatory determinants in maize kernels are shown [13,14,15], for a review refer to [14]. In dicots biosynthetic genes are commonly divided in Early Biosynthetic Genes (EBGs), which catalyse the early steps leading to the production of flavonoids from CHS to naringenin chalcone and related branches, and Late Biosynthetic Genes (LBGs) specific to anthocyanins and proanthocyanidins [16]. These structural genes are differently regulated. Indeed, the expression of the EBGs is modulated by a set of functionally redundant R2R3-MYBs, while the activation of LBGs is regulated by a ternary MBW complex [11]. In Figure 1b, the biosynthetic route and the major regulatory determinants of anthocyanin biosynthesis in tomato fruits are shown [17,18,19].

### 2.2. Carotenoids Colouring Fruits and Grains

Carotenoids are tetraterpenoids sharing a carbon chain structure with 9–11 conjugated double bonds. The ability of these pigments to absorb light of specific wavelengths is related to the length of the carotenoid chain, which in turn affects colouration in the range from yellow to red [20,21]. Unlike anthocyanins, carotenoids are synthesised in bacteria, algae, and plants [22]. The presence of carotenoids is particularly noticeable in fleshy fruits, such as tomato that accumulate a large amount of the red lycopene [23]. Accumulation of lycopene increases with ripening in tomato fruit and is considered as part of the ripening process [24]. In cereals, the carotenoid β-hydroxylases (HYD) reached the highest expression level at the last stage of development, suggesting that carotenoids (at least xanthophylls) are still actively synthesized in mature grains, raising the nutritional value of kernels [25]. Carotenoids are divided into carotenes (lycopene, α-, and β-carotene) and xanthophylls (lutein, and zeaxanthin) based on the absence or presence of oxygen, respectively, and their biosynthetic route is shown in Figure 2 [26,27,28,29,30,31,32,33,34].

## 3. Antioxidants in Cereals

Wheat, rice, and maize are the most important food crops and contribute more than 50% of calories of the human diet, while other cereals, e.g., sorghum and barley, are relevant only in some regions [35]. Due to their relevance in the diet, even a small increase in antioxidant content could have a large impact on health, but only if whole grain flour is used, since bran (pericarp) and germ are the major sources of dietary fibre, phenolics, vitamins, and minerals.

### 3.1. Wheat

In bread and durum wheat, the most important compounds with antioxidant properties, are phenolic acids (mainly ferulic and p-coumaric acids), and, to a lesser extent, carotenoids (mainly lutein), tocopherols, and tocotrienols (Table 1). Flavonoids also accumulate in wheat grains, mainly as apigenin derivatives [36]. Nevertheless, current evidence ruled out a significant antioxidant effect for this class of compounds [37]. Carotenoids in the form of lutein are the main determinants of the yellow colour of semolina, which has important implications for the marketing of durum wheat end-products [38], while apigenins contribute to the yellow colour of Asian alkaline noodles made from bread wheat [36]. Lastly, although rare in the currently cultivated wheat germplasm, anthocyanins provide the blue, purple, or red colour to the grains of pigmented wheats. Antioxidants have a specific tissue distribution into the grains. Most of the phenolic acids are contributed by outer layers of kernels (pericarp, and aleurone) or by germ [36,39], whereas the higher biomass of endosperm, compared to bran and germ, determine an overall low concentration of phenolic acids in kernels [39]. On the contrary, carotenoids are equally distributed across the kernel, with endosperm having the highest content [40]. The pattern of antioxidant content across kernel layers plays, therefore, a pivotal role for interpreting the variability of this trait in wheat germplasm. As antioxidants are mainly stored in the outer layers of mature grains, surface area to volume ratio of kernels is an important parameter that should be considered for exploiting the natural variation of this trait. Consequently, reducing grain size can improve the total antioxidant content in wholemeal flours at expenses of grain yield. Notably, not all phenolics present in wheat are equally bioavailable, they can be soluble free, soluble conjugated, or insoluble bound linked to polymers of the plant cell wall [41], with a per cent amount of 1, 22, and 77, respectively [42]. Soluble compounds have the highest bioavailability [42], with germination increasing soluble phenolics content and antioxidant activity [43], although some conjugated forms may also exert their biological activity upon cleavage by intestinal microbes [39]. To date, many studies have assessed natural variation in the total antioxidant content and its components in bread [44,45,46] and durum wheat (for phenolic acids [47,48]; for carotenoids and tocols [49,50]) considering varieties, landraces, and wild ancestors [51]. Some evidence might suggest higher amounts of phenolic acids in wild and some primitive subspecies, compared to durum wheat cultivars, and no significant differences between old and modern varieties [52,53], nevertheless these studies do not provide indication on the seed size and, consequently, the results cannot be considered conclusive. On the other hand, some publications, comparing historical and modern wheat varieties (e.g., [37,54]) suggested that breeding has qualitatively affected the profiles of phenolic compounds and isomer forms. Environmental factors (E, including year, location, as well as agronomic practices), genetic (G) effects, and GxE all contribute to determining phenotypic variation for total antioxidant contents [48,49,50,52,55], with environmental effects larger than genotypic differences [56]. Since plants produce secondary metabolites to cope with stress, both E and GxE effects are expected, the latter because of different levels of genetic tolerance to stress among genotypes. On the contrary, carotenoid content is largely controlled by genetic factors with a heritability value as high as 0.7 [57]. Consequently, breeding for yellow semolina colour has succeeded in a substantial quantitative effect on carotenoid content which is indeed significantly increased in modern varieties compared to landraces and old cultivars [49,58]. Two loci on durum wheat chromosomes 7A and 7B which co-localized with *PSY* genes [59], the rate limiting step of carotenoid biosynthesis [38], have a major effect on carotenoid content [60] and have been targets of successful phenotypic selection for yellow pigment content by recent breeding. As nutraceutical alternative, recent studies on durum wheat mutants in enzymes of the carotenoid biosynthesis provide support to successful engineering of the synthesis of β-carotene for the final goal of provitamin A biofortification of kernel durum wheat [61]. Grain pigmentation due to anthocyanin accumulation in pericarp or in aleurone can also represent a strategy to increased antioxidant activity in wheat. Two loci, *Pp3* and *Pp-1*, originally identified in a tetraploid Abyssinian wheat accession, are responsible for the activation of the anthocyanin biosynthesis pathway in wheat pericarp [62]. Purple pigmentation was initially used to clearly mark feed-type common wheat, but presently purple-grained commercial wheat cultivars for human nutrition have been released in several countries, e.g., the Canadian bread purple wheat AnthoGrain [63]. The genetic trait conferring blue aleurone has been introgressed in cultivated wheat from wheat relatives (*Agropyron* species and diploid wheat) [64] and then pyramidized with purple pericarp to obtain wheat line with a high anthocyanin content [65].

### 3.2. Rice

The most popular rice cultivars have a white pericarp, but variability for pericarp colour is well present in rice with accessions characterized by red, black, and purple pericarp mainly due to accumulation of proanthocyanidins that show a higher antioxidant content than white rice [81,82]. Two loci, namely *Rc* and *Rd*, encoding a bHLH transcription factor and DFR, respectively, control proanthocyanidin biosynthesis [83] and the accumulation of both anthocyanins and proanthocyanidins. Allelic variants in these loci account for a large fraction of the phenotypic variability for flavonoids and phenolic content [84,85], while additional loci and corresponding candidate genes controlling the content of ferulic acid have been identified [85]. In addition, *OsB2* and *OsC1* (*Kala4* and *Kala3*, respectively) have been proposed as responsible for anthocyanin accumulation in pericarp of black-grained rice [22,86]. A biotechnological approach has significantly contributed to increase the content of pigment with nutritional value in rice. The best-known example is Golden Rice. In rice endosperm of all existing wild and cultivated varieties, ß-carotene and other carotenoids have never been identified as the carotenoid biosynthetic pathway is blocked owing to lack of expression of *PSY*, *PDS*, and *ZDS* in rice endosperm [87,88]. Golden Rice is a genetically modified rice in which the biosynthetic pathway of carotenoids in the endosperm was restored by transforming plants with daffodil *PSY* and *crtI* from *Erwinia uredovora* under the control of an endosperm-specific or constitutive promoter, respectively [87]. Golden Rice plants carrying both transformation events showed 1.6 µg/g DW mean concentration of carotenoids in kernels, still insufficient to prevent diseases correlated with low carotenoid intake. As the limiting step of carotenoid biosynthesis in Golden Rice was recognized to be the daffodil *PSY* gene, a panel of different heterologous *PSY* genes was tested and the combination between maize *PSY* and *E. uredovora crtI* genes resulted in a carotenoid content up to 37 µg/g DW (Golden Rice 2, [89]), making this genetically modified rice useful to fight carotenoid malnutrition. Bioavailability studies demonstrated that 100–150 g of cooked Golden Rice 2 (50 g uncooked) provides about 435 µg retinol and is as effective as β-carotene supplementation in oil capsules in providing vitamin A to 6–8 years-old Chinese children, reaching 60% of Chinese Recommended Daily Allowance of vitamin A [90].

### 3.3. Maize

Maize is naturally rich in antioxidants and pigmented varieties of corn could be used as healthy ingredients for the formulation of novel gluten-free products [91]. The antioxidant activity of maize flour is dependent on phenolics, mainly ferulic acid, carotenoids, phlobaphenes, and anthocyanins, which confer different colours to five types of corn (white, yellow, blue, purple, and red). The composition in antioxidants and bio-active compounds (phenolics, proanthocyanidins, carotenoids, and tocols) was recently found to vary in differently pigmented corns [92]. The total phenolic content ranges from 2.60 to 17.56 mg of gallic acid equivalents (GAEs)/g with the lowest content in white corn and the highest in Andean purple corn, whereas yellow corn contains 3.20 mg of (GAEs)/g [73,74]. Phenolic acids are mainly represented by ferulic acid and p-coumaric acid, particularly in pigmented varieties [73]. In maize seeds, anthocyanin accumulation is genetically determined by *MYB* and *bHLH* regulatory genes and can occur in the aleurone (blue corn, *C1 R1*), in the pericarp (purple corn, *B1 Pl1*) or, eventually, in both, with an anthocyanin content ranging between 0.66 and 1.64 mg of cyanidin 3-glucoside equivalents/g [77]. An alternative source of anthocyanins suitable for the development of dietary supplements is the purple corn cob with a concentration range of 3.1 to 12.6 mg of cyanidin-3-glucoside equivalents/g [79,80]. Finally, red pigmentation depends on the *MYB P1* gene, which activates phlobaphene accumulation (27.53 mg/g) in pericarp of maize seeds [91]. Yellow corn varieties contain lutein, zeaxanthin, β-cryptoxanthin, and β-carotenes mainly in the germ, followed by aleurone and endosperm [68,93,94]. Carotenoid concentration can also vary depending on genotypes, ranging from 0.18 μg of β-carotene/g in the blue corn [69] up to 60 μg of xanthophylls/g in high carotenoid varieties [95]. Several synthetic anthocyanin-rich varieties suitable for human consumption (i.e., polenta, sweet corn, and popcorn) have been selected aiming to increase the daily intake of these bioactive compounds [96,97] and when tested the higher scavenging ability of anthocyanin-rich varieties compared to control yellow varieties was confirmed [13].

### 3.4. Barley

Barley has only a limited use for direct human consumption, especially in the form of hull-less (naked) grains. Whole grain barley was shown to reduce the risk of developing chronic diseases thanks to health-promoting properties of different bioactive compounds, such as β-glucan fibers and antioxidants [98]. Barley is richer in vitamin E (α-tocopherol and α-tocotrienol) than most cereals and contains high levels of free or conjugated phenolic compounds with antioxidant properties ([99], Table 1). In addition, barley genotypes can accumulate anthocyanins in different seed external layers resulting in yellow, blue, purple, or black barley grains. The *Ant28* locus on chromosome 3H controls the biosynthesis of proanthocyanidins, that accumulate in the seed coat giving a yellow colour [95]. Two MYB transcription factors on chromosome 4H are the best candidate genes for the accumulation of anthocyanins in the aleurone layer, that results in blue grains [100]. On the other side, the accumulation of anthocyanins in the pericarp, controlled by *Ant1* and *Ant2* loci that determine the red pigmentation of awns and auricles, results in purple grains [101]. The heterogeneous distribution of these compounds can be exploited through pearling, to separate fractions enriched in specific compounds. For example, Martínez-Subirà et al. [102] have recently observed that the outermost 30% pearling fractions best exploit barley antioxidant capacity and could be used as a valuable source of functional ingredients. Biscuits containing different proportions of whole flour and pearling fractions from a purple hull-less barley genotype showed high in vitro antioxidant capacity [103]. Interestingly, genotypic differences influence the content of bioactive compounds more than the environment, and heat stress at the end of the growing season increased the phenolic compounds and their antioxidant capacity [104].

### 3.5. Sorghum

Sorghum grains, particularly red varieties, exhibit the highest values of total antioxidant capacity (400–500 μmol of Trolox equiv/g) among several crops (e.g., wheat, rice, oats, barley, maize, and potato) [105,106] and sources of natural antioxidants from plant foods [107,108]. Polyphenols are the major component of antioxidant capacity in sorghum grain and several studies [75,78,109] showed broad genetic diversity for their content, implying that breeding activity can improve antioxidant capacity in sorghum crop. For instance, Habyarimana et al. [109] and Salazar-López et al. [75] came across similar ranges of phenolic contents of, respectively, 0.6–20.73 and 1.0–29.6 mg equivalent of gallic acid/g (mg GAE/g) in diverse sorghum populations, with wild relatives outperforming the cultivated species. Polyphenols, particularly tannins, were long selected against in sorghum [108], because of their astringency and negative effect on animal feeding. As human interest shifted in favour of these compounds, it can be expected that such a trend is going to be reversed with wild relatives representing useful traits donors [110]. Quantitative traits loci for antioxidant content, grain colour, and antioxidant capacity were reported in sorghum [111] and two major genes were proved to be involved in the biosynthesis of polyphenols in grain [108,112]. They encode a MYB (Yellow seed1) and a WD40 (Tannin1) transcription factor, the latter having homology to the *A. thaliana* TTG1. Several other genes have been associated to variations in grain tannin and antioxidant activity, including two homologs of *AtTT10* and *AtTT4* [113,114].

## 4. Antioxidants in *Solanaceae*

The *Solanaceae* family comprises economically important staple crops, like tomato, potato, eggplant, and pepper. Millennia of domestication and breeding have selected many varieties, in which the levels and distribution of carotenoid and anthocyanin pigments vary widely in response to developmental and environmental stimuli (Table 2). Their accumulation in fruits is an important determinant of ripeness and quality and is frequently used to easily differentiate the products.

### 4.1. Tomato

Tomato displays a plethora of pigmented phenotypes; nevertheless, conventional commercial varieties exhibit red pigmented fruits due to the presence of lycopene [115]. Cultivated tomato fruits are generally devoid of anthocyanins [116] and the current black varieties are due to chlorophylls/carotenoids co-pigmentation [117]. The loss of anthocyanins occurred during the first wave of domestication [118], although under favourable environmental conditions (e.g., high light and/or cold temperatures), anthocyanin pigmentation can eventually be observed [119]. Indeed, the anthocyanin biosynthetic machinery is fully present in the fruits of cultivated tomato, but inactive due to mutations that occurred in regulatory players thus leading to the red fruits phenotype [120] for a recent review see [121]. The anthocyanin trait was reintroduced in modern tomato cultivars through the introgression of the *Aft* [18,122] and the *atv* genes [123,124] derived from *S. chilense* and *S. cheesmaniae* (L. Riley) Fosberg wild relative tomatoes, respectively. The resulting *Aft*/*Aft atv*/*atv* tomato line, commercially known as Sun Black™, shows the anthocyanin pigmentation limited to fruit skin [125]. Interestingly, total carotenoids content is not affected [126]. Using the same parental lines and breeding program of Sun Black™, the University of Oregon developed an independent variety, named “Indigo Rose”, with the same phenotype (https://extension.oregonstate.edu/news/purple-tomato-debuts-indigo-rose). Many biotechnological strategies have been exploited leading to a significant enrichment in lycopene, anthocyanins, and other flavonoids. Silencing the *SGR1* gene via genome editing promoted the activity of *PSY1* during fruit maturation, which increased the accumulation of lycopene (up to 5-fold) and β-carotene [127]. The pyramiding of mutations in four genes controlling the carotenoid biosynthesis allowed the accumulation of zeaxanthin, a carotenoid with high nutritional value absent in tomato fruits. The BSh allele of the *CycB (B)* gene (identified in wild accessions) promotes the accumulation of β-carotene up to 80% of total carotenoids. When *BS^h^* was introgressed in genotypes carrying *high-pigment 3* (*hp^3^*, a mutation impairing in zeaxanthin epoxidase) [128], *GREEN-STRIPE* (gs, a mutation conferring green fruits) and high-pigment mutant *hp2^dg^* [129], the quadruple homozygous mutant *BS^h^ hp^3^ gs hp2^dg^* accumulated >500 µg/g DW of zeaxanthin in fruits [130]. Metabolic engineering allows to accumulate anthocyanins up to 2 mg/g FW in *Del/Ros1* purple tomatoes, a level comparable to small berries [120] as well as to drive the selective accumulation of other flavonoids. Indeed, the Cathie Martin’s research group developed several transgenic tomato lines enriched in polyphenols (Table 3); among them, the “Bronze” fruits showed the highest antioxidant capacity compared to both red and other polyphenol-enriched lines due to the simultaneous presence of different classes of polyphenols (flavonols, anthocyanins, and stilbenoids) [131,132].

**Table 2 antioxidants-11-00794-t002:** Typical range of concentration of phytochemicals in the edible fruit of *Solanaceae* varieties. * In eggplant fruits commercial maturity (stage B) precedes the physiological ripening (stage C). n.d. = not detectable.

				Typical Range of Concentration (mg/g) in the Edible Fruit of Cultivated *Solanaceae* Varieties			
Antioxidant Class	Prevalent Compounds	Main Modifications	Distribution into the Fruit	Tomato (*S. lycopersicum*)	Potato (*S. tuberosum*)	* Eggplant (*S. melongena*)	Pepper (*C. annum*)
Carotenoids	lycopene and phytoene (tomato); antheraxanthin, violaxanthin, lutein; zeaxanthin (potato); luteolin (pepper)	n.d.	peel and flesh	Cultivated varieties (mg 100 g^−1^ FW) 7.8–18.1 (lycopene); 1.0–2.9 (phytoene) [133]	Cultivated varieties (mg/100 g DW): 1.1 (antheraxanthin); 0.8 (violaxanthin); 0.5 (lutein); 0.5 (zeaxanthin in *S. tuberosum* 4n); 2.2 (zeaxanthin in *S. phureja* 2n) [134], 0.3–3.6 (mg/100 g DW) in Andean landraces [135]	Local eggplant landraces 0.00146–0.00406 mg g^−1^ FW [136]	Red cultivars total carotenoids 13.51–43.32 mg/100 g DW; orange cultivars total carotenoids 109.69–190.43 mg/100 g DW; yellow cultivars total carotenoids 15.31–29.70 mg/100 g DW [137,138]
Phenolic acids	hydroxycinnamic acids (mainly CGA, caffeic acid)	mainly conjugated with organic acids	mainly in the flesh where they account for 70–90% of total phenolics in eggplant fruits; mainly in tuber skin (potato)	Cultivated varieties (mg 100 g^−1^ FW) 1.4–3.3 (CGA); 0.1–1.3 (caffeic acid) [133]	Cultivated varieties (tuber skin) 1020–2920 mg/100 g DW, CGA: 2.11 mg g^−1^ DW (cv. Bionica) [139,140]; total phenolics: 10.1–105.0 mg/100 g DW (Vitellotte), 6.7–108.0 mg/100 g DW (Blue Star) [141]	Cultivated eggplant CGA at stage A of fruit ripening (2319 mg/100 g DW) [142], 4 mg/100 g DW cv. Lunga Napoletana [143]	*C. annuum* cultivars 119.97 ± 3.44–2060.12 ± 20.56 mg GAE/100 g FW [127]
Flavonoids (considered as total content)	flavanones (naringenin chalcone), flavonols (mainly rutin, quercetin)	mainly conjugated (glycosides)	mainly in the fruit peel	Cultivated varieties (mg 100 g^−1^ FW) 0.9–18.2 (naringenin chalcone); 0.5–4.5 (rutin); 0.7–4.4 (quercetin) [133]	Cultivated varieties (tuber skin): total flavonoid content 510–960 mg/100 g DW [139], total flavonols 0–0,22 (Andean potatoes) and up to 3 (Phureja tubers) mg g^−1^ DW [144]	Cultivated varieties at commercial maturity (stage B) total flavonoid content 267.7 mg/100 g DW [145]	Cultivated varieties (mg 100 g^−1^ FW): 2.21 (quercetin); 4.71 (luteolin) (USDA)
Flavonoids (Anthocyanins)	mainly delphinidin, petunidin (potato), and pelargonidin-3-(p-coumaroyl-rutinoside9-5-glucoside	mainly conjugated (glycosides)	mainly in the fruit peel	n.d.	Anthocyanins: 16 (Andigenum group) and 41 (Phureja genotype) mg g^−1^ DW [144], 16.32 mg/100 g DW (Vitellotte), 18.12 mg/100 g DW (Magenta Love) [141]	cv. Lunga Napoletana D3R ∼1.2 mg/100 mg DW [143]	n.d.

### 4.2. Potato

Cultivated potato varieties show a great variability in terms of accumulation of diverse xanthophylls in the tubers flesh and skin [153], ranging from 2 (white-fleshed) to 45 (yellow fleshed) mg/kg of FW [154]. Allelic variation in several genes involved in the biosynthesis of the carotenoids is responsible for the genetic diversity in tuber colour. Two alleles at the *BCH2* gene [155] (Figure 2) explaining a large variation in carotenoid content are currently used in breeding [156] with carotenoid accumulation in tubers correlated with high *CHY2* expression level [157,158]. The carotenoid profile of most diploid Andean native accessions of the Phureja group (collectively called papa amarilla, due to their yellow-orange tuber flesh colour) is characterized by a marked zeaxanthin accumulation, that can reach up to 85% of the total carotenoids in the tuber, while in Tuberosum group it seldom exceeds 24% [134]. Nevertheless, due to ploidy incompatibility, this trait cannot be easily transferred can be hardly transferred by conventional breeding in tetraploid cultivated varieties. The phenotype “high-zeaxanthin” is sustained by the presence of a defective allele at the *ZEP* locus (Figure 2) that impairs the synthesis of antheraxanthin and downstream carotenoids resulting in zeaxanthin accumulation, conferring the typical orange colour of the tuber flesh [157,158]. Cultivated potato varieties can also accumulate anthocyanins, mainly delphinidin- and pelargonidin-based molecules [10], conferring red or purple colour, while the Andean potato tubers show a wide variability in flesh and skin colour mainly due to a combination of anthocyanins and carotenoids [158]. Targeted genetic manipulation to improve nutritional quality of potato tubers has been accomplished by biotechnological approaches aiming at enhancing zeaxanthin content as well as other carotenoids. Of relevance, zeaxanthin concentration in tubers was tripled by silencing the *LCY-E* gene [149] and was increased up to >5 fold by *ZEP* gene silencing [150], with a single average-sized potato contributing about 0.5–0.6 mg of zeaxanthin in diet.

### 4.3. Eggplant

The pigmentation of eggplant fruits is associated to the presence of two anthocyanins: delphinidin-3-rutinoside (D3R) and nasunin (NAS), which accumulate at high amounts in the peel of black-purple and lilac genotypes, respectively, due to the activity of an acyltransferase [159]. β-carotene is only marginally present in eggplant fruits, but the transgenic expression of the *crtB* gene under the control of the fruit specific *EEF48* promoter allowed to increase β-carotene content up to 30-fold compared to WT [158]. Cholorogenic acid (CGA) is the major soluble phenolic compound in eggplant predominantly accumulated in fruit flesh [141]. In cultivated eggplant varieties and in their wild relatives (i.e., *S. incanum*) a large genetic diversity for CGA content has been described, thus allowing breeders to introgress the high-CGA trait into commercial cultivars and to counter-select for low polyphenol oxidases (PPO) activity to limit the browning pleiotropic phenotype [160]. Recently, a gene editing strategy targeting the genes coding for PPO allowed to select genotypes enriched in beneficial phenolic compounds without undesired browning phenotypes [161].

### 4.4. Pepper

Peppers are characterized by a wide range of colours (varying from red to white), phenolics and carotenoid profiles [138], and pungency. Nevertheless, the molecular identity sustaining the different pigmentations is known only for the orange colour attributed to a splicing mutation in the *ZEP* gene leading to the accumulation of zeaxanthin [162]. Fruits consumed at the at the green stage are deficient in antioxidant content, due to the progressive accumulation of carotenoids, phenolics, flavonoids, and ascorbic acid in later stages of ripening [163]. Indeed, green pepper fruits have a total carotenoid content of the order of magnitude of 10–50 mg/100 g DW, while at full maturation carotenoid content can peak as high as 3000 mg/kg DW (e.g., in mature red paprika; [164]). The peculiar carotenoids of red peppers are capsanthin, capsorubin, and capsolutein [165]. Red fruits mainly contain xanthophylls including ketoxanthophylls and epoxydated forms of these carotenoids. Pepper carotenoids occur in esterified form, with fatty acids that make them insoluble in mature fruits [164]. Anthocyanin pigmentation is not appreciated by breeders, because restricted to unripe pepper fruits, thus determining a bitter taste and lack of sweetness [10].

## 5. Impact of Crop Management and Food Processing on Antioxidant Content

Besides genetic factors, different agronomic practices (e.g., fertilization), management (conventional vs. organic) and symbiotic association with arbuscular mycorrhizal fungi can affect polyphenols biosynthetic pathway [166]. In general, higher levels of flavonoids and antioxidants are observed at low nitrogen input [167], for instance, tomato fruits harvested from plants grown with low nitrogen supply tend to have a higher phenolic content [168]. Nevertheless, some exceptions are also reported, and anthocyanin accumulation was found higher in purple-blue potatoes fertilized with 120 kg ha^−1^ of nitrogen in the absence of phosphorus and potassium fertilization [169]. The effect of crop management (organic vs. conventional practices) cannot be univocally explained. A long-term study of tomatoes showed that total phenolics and flavonoids, in particular quercetin, naringenin, and kaempferol content, were higher in fruits from organic plots, a result associated to the reduced amount of available nitrogen in the organic management [170]. Potatoes from organic cultivation were significantly richer in polyphenols, including phenolic acids, and flavonoids, in comparison to the conventional ones, while higher concentrations of lutein were noted in conventional compared to the organic potatoes [171,172]. However, results are often contradictory due to the strong influence of both environment and genotype, suggesting that the beneficial effects of organic farming can be maximized by using new varieties developed for a low-input agriculture [173,174]. Oat [174] and rice [175] grains as well as eggplant fruits [176] exhibited no differences in the polyphenols content between organic and conventional farming. In some studies, higher soluble phenols and total flavonoids were found in organic pepper as compared to conventional one, in response both to pathogenic pressure and to a lower nitrogen availability [177], while in other works no differences were pointed out [178,179]. In response to higher incidence/severity of pest and disease damage, due to the ban of synthetic pesticides in organic farming and/or fertilization intensity, organic cultivation may lead to an intensive production of secondary metabolites, including those with antioxidant properties, as a defence mechanism [174]. Research on microbial communities that colonize plant surfaces (phyllosphere, and rhizosphere) and internal plant tissues (endosphere) has shown that these microorganisms can also modulate the plant secondary metabolism [180], leading to increased biosynthesis of antioxidant phytochemicals in response to stress [181,182]. Studies on crops treated with arbuscular mycorrhizal fungi have evidenced that secondary metabolites and antioxidant activities varied depending on the variety and on the composition of the mycorrhize inoculum, ranging from no effect to significantly increased antioxidant concentrations [183]. Mixes of *Gomus* species resulted efficient in increasing antioxidants in tomato (up to 18.5% lycopene) in controlled conditions (greenhouse) [184].

Cereals and *Solanaceae* often undergo transformation and/or cooking before being eaten and these processing heavily impact on the antioxidant content of foods [10,183]. In cereals, a first significant decay of bioactive compounds is observed after traditional milling with a significant amount of antioxidant lost with the bran [185,186]. Strategies to enhance the antioxidant potential of flours are the emerging techniques based on alternative milling methods (micronization, and microfluidization) [187,188] or dedicated fractionation processes (debranning/pearling and air-classification) to select fractions rich in phytochemicals of the external grain layers [182,189,190]. The level of bioactive compounds is further affected by processing and cooking [191]. In bread-making, major decreases in carotenoids and anthocyanins were shown during kneading and baking phases, respectively [192,193]. In pasta-making, the critical step was represented by the drying process, with a up to half decay in anthocyanins of its original content (in pasta made with purple durum wheat) [183]. Significant is also the partial leaching of bioactive compounds during pasta cooking [183,194]. Quite different is the impact of cooking on rice prepared following the Italian traditional “risotto”-like cooking style, in which during boiling, broth or water is completely reabsorbed by rice grains, causing a limited loss of anthocyanins with respect to carotenoids because of their different solubility [195].

In *Solanaceae*, after the preliminary preparation procedures (cutting and/or crushing, homogenization, and peeling) which are responsible for the enzymatic oxidation processes with a partial decay of antioxidants, the critical step is represented by thermal processing [191,196,197,198]. In tomato, during processing to produce tomato paste from fresh tomatoes, thermal treatment causes the breaking of protein aggregates where the pigment is associated inside the vegetable matrix, enhancing the carotenoid bioaccessibility [191]. A 3- to 4-fold increase of anthocyanins was found in potato after heating due to the enhanced chemical extractability of anthocyanins and depending on the variety used and the pH used in the extraction process [199]. Besides thermal treatments, other compounds of the food matrix affect the anthocyanin stability, such as metal ions, protein, fibers, and interaction with other phytochemicals [200,201]. In eggplants, thermal treatments lead to a decrease in bioactive compounds, with effects directly correlated with the increasing temperature [202,203]. On the contrary, frying or boiling potatoes may lead to important losses of carotenoids [201]. In purple eggplant, boiling caused more anthocyanin decrease than steaming [201]. A similar phenomenon has been reported for total anthocyanin of purple-fleshed potatoes boiled in 100 ◦C water with respect to steamed potatoes [203]. Innovative non-thermal treatments can be applied to protect bioactive compounds during food processing. High hydrostatic pressure processing was positively applied in tomato purees for preserving the bioactive compounds [204,205], while in chili pepper high hydrostatic pressure combined with a mild thermal treatment increased preservation and bio-accessibility of pigments [181,206].

## 6. Impact of the Cereal and *Solanaceae*-Based Foods Enriched in Antioxidants on Human Health: Preclinical and Clinical Studies

Although the in vitro antioxidant activity of plant polyphenols is higher than vitamins E and C [207], their low bioavailability and their conversion into metabolites by the gut microbiota and endogenous metabolism [208,209,210] suggest that polyphenols mainly exert their direct role of scavengers in the gut, whereas at cellular level they can function as molecular signals, able to activate the endogenous antioxidant defences [211,212]. Several studies have shown that polyphenols can activate the NRF2 factor and genes encoding for antioxidant response enzymes, such as superoxide dismutase, catalase, and glutathione peroxidase, but they can also directly modulate their enzymatic activity [212,213]. Polyphenols are also well-known for their anti-inflammatory properties by modulating both cellular and molecular mediators of inflammation [214,215]. Regarding the anti-inflammatory effect, several studies have established that polyphenols can prevent excessive and/or chronic inflammation by reducing the activation of NF-κB, a transcription factor that regulates many genes of the inflammatory response, such as *iNOS*, *COX-2*, and pro-inflammatory cytokines, such as IL-6 and TNF-α [216]. They also inhibit the signaling cascade of MAPKs (i.e., p38, JNK, and ERK), again reducing the activation of pro-inflammatory cytokines, iNOS and COX-2 [217] and directly inhibit COX-1 and COX-2 enzymes thus reducing the production of prostaglandins E2 (PGE2) [218].

Taken together, the in vitro data indicate an overall bioactive potential of plant antioxidants for the prevention of oxidative stress and chronic inflammation, nevertheless the effectiveness of antioxidant-rich foods has also to be tested in preclinical studies and validated with human clinical trials.

### 6.1. Pre-Clinical Studies

Several in vivo studies indicate that supplementation with pigmented rice and wheat may contribute preventing metabolic syndrome by ameliorating the negative effects of high-fat or high-sugar diets. Table 4 summarizes the pre-clinical studies which highlight the health effects of antioxidant-rich cereals and *Solanaceae* foods. In mice models of diet-induced obesity, black rice was more effective than simvastatin in decreasing cholesterol metabolism and prevented hypertriglyceridemia, weight gain, insulin resistance with an improvement on gut microbiota dysbiosis [219,220,221,222]. Both red and black rice decrease atherosclerotic plaques in rabbits and increase the endogenous antioxidant status in apoliprotein E (apoE)-deficient mice [223,224,225]. Similarly, both black and purple wheat reduced total cholesterol, triglyceride and free fatty acid levels in serum, with restoration of insulin resistance and elevation of antioxidant enzymes [226]. Black rice also exerted a protective action against alcohol-induced liver toxicity and hepatic fibrosis induced by carbon tetrachloride as demonstrated by the decreased plasma levels of aspartate transaminase, alanine transaminase, and gamma glutamyl transferase [227,228]. Finally, an anti-aging effect of black rice was shown in a mouse model of D-galactose-induced senescence, by increasing the endogenous antioxidant response [229] and a neuroprotective effect of purple rice was suggested in rat models of Alzheimer’s disease, by preventing memory impairment and hippocampal neurodegeneration [230]. A wheat-based anthocyanin-rich diet also prevented deficits in working memory in a mouse model of Alzheimer’s disease induced by amyloid-beta and partially reversed episodic memory alterations compared to a diet based on non-pigmented wheat. Furthermore, pigmented wheat reduced alpha-synuclein accumulation in a mouse model of Parkinson’s disease and enhanced the expression of Arginase1 that marks M2 anti-inflammatory microglia in the brain [231]. Evidence from animal studies also support the beneficial effect of polyphenol-rich corn on a variety of chronic diseases, such as cardiovascular disease, obesity, diabetes, cancer, and some genetic diseases [215,232]. Dietary treatment with anthocyanin-rich corn was able to induce a long-lasting cardioprotection against ischemia-reperfusion injury by modulating the microbiota [233,234] and to prevent the cardiotoxic effects induced by doxorubicin, a chemotherapeutic drug widely used against solid and haematological tumors [235]. Several in vivo studies have shown the ability of purple corn to ameliorate hyperglycemia, insulin sensitivity and to retard diabetic nephropathy in diabetic *db* mice [236,237,238], to prevent high fat diet-induced obesity as well as to reduce obesity-associated inflammation [77,239,240,241]. Purple corn was also effective in reducing the development of trigeminal inflammatory pain in rats [242] and in preserving muscle function in a dystrophic mouse model, in which oxidative stress and inflammation trigger progressive muscle tissue loss [243].

Pigmented-rich tomato or potato have a positive impact on health in animal studies. In mice, in vivo studies with polyphenol-rich tomato have suggested potential antitumoral and anti-inflammatory activities. Mice fed with an anthocyanin-rich diet from *Del*/*Ros1* tomato had significantly prolonged lifespan in cancer-prone *p53^−^*^/*−*^ mice [120] compared to red tomato. An extract from a V118 purple tomato significantly reduced carrageenan-induced paw oedema in rats in a dose-dependent manner compared to red conventional tomato, supporting a strong anti-inflammatory activity of purple tomato, associated to a reduction of oxidative stress, as evidenced by reduced lipid peroxidation, nitric oxide production, and increase of detoxifying enzymes [244]. Tomlinson et al. [245] reported strong and specific inhibitory effects on gut barrier pro-inflammatory cytokines and chemokines via SAPK/JNK and p38 MAPK pathways of engineered tomato extracts (enriched in anthocyanins from *Del*/*Ros1* tomato or enriched in flavonols from *AtMYB12*-overexpressing tomato) by primary murine colonic epithelial cell-based inflammation assays. Based on this study, tomato lines with different combinations of polyphenols (Table 3) have been tested in a mouse model of inflammatory bowel disease (IBD), demonstrating that tomato enriched in flavonols, anthocyanins, and stilbenoids (named Bronze) were able to reduce/delay the symptoms as well as the dysbiotic intestinal microbiota associated to dextran sodium sulfate (DSS)-induced colitis, and showed a significantly diminished IL-6 and TNF-α levels. Interestingly, the combination of different polyphenols was more effective than single flavonoid classes [132,246]. A similar preventive effect on inflammatory cytokines and gut barrier function was observed using purple-fleshed potato on high-fat diet in pigs and on DSS-induced colitis in mice [247,248,249]. Purple and red potatoes reduced benzopyrene-induced stomach cancer and chemically induced breast cancer in rats [250,251]. Purple potato showed an anti-obesity effect in rats or mice and high-fat diet through p38 signaling pathway and UCP3 [252,253]. In addition, purple potatoes were reported to reduce hyperglycemia, oxidative stress, and overall food intake in diabetic rats [254] and to increase the antioxidant response pathway in hypercolesterolemic rats [255,256,257].

### 6.2. Clinical Studies

The in vitro and animal studies provide some evidence that antioxidants are beneficial for health and, consequently, the increase in antioxidants content has been pursued particularly in cereals and *Solanaceae*, since they represent relevant dietary sources of nutrients in the human diet. A plethora of scientific evidence, summarized in Table 5, suggests that increased consumption of fruits, vegetables and cereals represents an easy and practical strategy to significantly reduce the incidence of chronic diseases, such as cancer, cardiovascular diseases, type-2 diabetes, obesity, and other aging-related pathologies [250,251,252,253,254,255,256,257,258,259,260]. A large population-based study (i.e., 24,325 men and women aged ≥35 years) has evidenced that a high polyphenol antioxidant content (PAC) score in the diet [261] is associated in both genders with low-grade inflammation, evaluated as C-Reactive Protein levels, white blood cells, platelet count, and granulocyte to lymphocyte ratio [262], with a reduced mortality risk, including cerebrovascular and cancer mortality [263], and with a decelerated biological aging [264]. In addition, a meta-analysis including many randomized controlled trials reported that the consumption of polyphenol-rich berries significantly ameliorated lipid profiles, blood pressure, or vascular functions [265]. Nevertheless, the potential impact of antioxidant-rich cultivars in comparison with conventional cultivars in the context of a generally healthy diet have been studied so far in very few clinical studies and deserves further investigation. Much evidence supports the potential beneficial properties of whole grain cereals, a result associated to their unique composition in vitamins and minerals, unsaturated fatty acids, tocotrienols, tocopherols, soluble and insoluble fiber, phytosterols, stanols, sphingolipids, phytates, lignans, and lipophilic and hydrophilic antioxidants, such as phenolic acids [46,259,266,267,268]. The synergistic effect of these compounds, along with other nutrients, is responsible for health benefits associated to whole grain cereals [269]. Nutritional guidelines suggest increasing whole-grain intake by replacing refined grains with whole grains [270,271]. Many works have been concerned with the antioxidant properties, chemistry, and functions of coloured rice, wheat bran [272], and pigmented maize, nevertheless only very few studies in humans are available in which antioxidant-enriched grains are compared vs. control varieties. A cross-over acute study conducted to evaluate the effect of bread made from the anthocyanin-rich rice cultivar “Riceberry” on the postprandial glycemic response, glucagon-like peptide-1, antioxidant status, and subjective ratings of appetite on 16 volunteers, suggest a reduced glycemic response together with improvement of antioxidant status in healthy subjects [273]. A cross-over design, randomized, dietary intervention on 24 healthy subjects was carried out by Callcott [274] to assess the impact of pigmented rice (purple, red, and brown) consumption on antioxidant status and reduction of biomarkers of oxidative stress and inflammation. This study demonstrates that acute consumption of red and purple rice in a healthy population significantly increases antioxidant activity and decreases plasma MDA and proinflammatory cytokines. The brown rice variety did not affect any parameter tested. A clinical pilot study showed that consumption of anthocyanins from purple corn reduced plasma levels of inflammatory markers and improved the response to Infliximab, a chimeric monoclonal antibody against TNF-α, in Crohn’s Disease patients [275]. Liu and co-workers [276] in a clinical trial on 120 individuals affected with type-2 diabetes found that black wheat intake (69 g/d) for 5 w decreased serum levels of glycated albumin and prevented the increase in TNF-α and IL-6 levels compared with the control group, in which the subjects only received nutritional education and diet control. Although studies with larger sample sizes or longer durations are needed, the authors concluded the partial substitution of black wheat can alleviate or prevent hyperglycemia and inflammation in type-2 diabetes patients or in individuals at high risk of developing type-2 diabetes. In a randomized, single-blind parallel-arm study, Gamel et al. [277] found a modest improvement in plasma markers of inflammation and oxidative stress in overweight and obese adults with evidence of chronic inflammation (high-sensitivity C-reactive protein > 10 mg/L) upon consumption of both purple and regular wheat.

While in cereal-based foods a relevant increase in the content of beneficial compounds can be better achieved moving from refined to whole grain products, tomato fruits and tomato-based products are both good sources of lycopene as well as other antioxidant components, including ascorbic acid and phenolic compounds. Recent genetic achievements have led to the selection of anthocyanin-rich tomato varieties that could significantly increase the beneficial effect of tomato on human health. Considering the promising results obtained in some in vitro and in vivo preclinical studies investigating the health potential of purple tomato, human studies aimed to validate these findings deserve to be undertaken [121].

Clinical studies showed beneficial effects of pigmented potato consumption in healthy adults through reduced inflammation and DNA damage [278], significant drop in blood pressure without weight gain [279] and reduced postprandial glycemia and insulinemia [280,281]. A small-scale intervention study has shown that purple potatoes improved arterial stiffness in healthy adult [282]. Eggplant rich of bioactive compounds, such as anthocyanins and phenolic acids, has been extensively investigated for its antioxidant properties, for a strong radical scavenging activity [283,284]. Nasunin, the major component of anthocyanin pigment of eggplant, has a potential bioactive role in the control of osteoblastic for bone health since in vitro studies have shown that nasunin prevents MC3T3-E1 osteoblast cells from oxidative damage [285]. It remains to be clarified whether the use of nasunin as a functional food ingredient in the diet may be relevant for human bone health depending on their bioavailability after ingestion [286]. Although the results of previous in vivo studies are promising, as they show that nasunin given to rats as a food supplement is well absorbed from the gastrointestinal tract [287], clinical trials to evaluate the effects of dietary nasunin on bone mass are lacking [283].

**Table 5 antioxidants-11-00794-t005:** Health effects of antioxidant-rich foods in clinical studies. MDA, malondialdehyde; HCD, high-caloric diet; HCD, TNF-α,Tumor Necrosis Factor-α; and DNA, deoxyribonucleic acid.

	Antioxidant-Rich Food	Health Effects in Clinical Studies	Refs.
*Cereals*	Antocyanin-rich rice	A reduced glycaemic response together with improvement of antioxidant status in healthy subjects	[273]
	Purple/red rice	Significantly increases antioxidant activity and decreases plasma MDA and proinflammatory cytokines in healthy population	[274]
	Purple corn	Reduced plasma levels of inflammatory markers and improved the response to Infliximab, a chimeric monoclonal antibody against TNF-α, in Crohn’s Disease patients	[275]
	Pigmented wheat	Various medical advantages like obesity, type-2diabetes, cardiovascular disease, and cancer	[272]
	Black wheat	Decreased serum levels of glycated albumin and prevented the increase in TNF-α and IL-6 levels in patients with type 2 diabetes	[276]
	Purple wheat	Modest improvement in plasma markers of inflammation and oxidative stress in overweight and obese adults with evidence of chronic inflammation	[277]
*Solanaceae*	Pigmented potatoes	Significant drop in blood pressure without weight gain in healthy adults	[279]
		Reduced postprandial glycemia and insulinemia in healthy adults	[280,281]
	Purple potatoes	Improved arterial stiffness in healthy adult	[282]

## 7. New Perspectives to Promote Antioxidant Response in Human Cells: Plant miRNAs with Cross-Kingdom Activity

Plant-derived foods also contain a class of small RNA that to date, only a few investigations have been performed for their antioxidant and beneficial effect on human health. Very recently, it has been reported that microRNAs (miRNAs) can transfer and regulate gene expression in a cross-kingdom manner [288]. Stable miRNAs derived from food plants may enter the mammals’ circulatory system and inhibit the production of target mammalian proteins These observations have been received with a mix of enthusiasm and scepticism [289]. This new type of regulatory mechanism, although not well understood, provides a fresh look at the relationship between food consumption and physiology [290]. For example, miRNA designed based on *Fragaria vesca* miR168, can reduce inflammation and prevent symptoms of multiple sclerosis at physiological concentrations [291]. The mechanism causing immune-modulating activity of plant miRNAs is poorly understood and may be due to the 2′-OH-methylation at the 3′ end and to the capacity of miRNAs to bind to toll-like receptor 3 (TLR3) and impairing TIR-domain-containing adapter-inducing IFN-β signalling. The orthologous from rice, osa-miRNA168, reduced T cell proliferation to a similar extent, thus suggesting the existence of analogous mechanisms in mono and dicotyledonous species [291]. This is also supported by the miRNAs found in human blood, which are mapping to corn, rice, and wheat [292]. There is scope for the breeding of crops with increased amount of miRNAs, as demonstrated by the similarities and differences in expression profiles between tissues of the same genotype as well as between genotypes, for example in alfalfa [293] and maize [294]. Also, it is possible to envision agronomic systems designed to optimize miRNAs biosynthesis [295]. Plant exosome-like nanoparticles can be taken up by immune system cells and exert antioxidant and anti-inflammatory effects [296]. As the exosomes which are absorbed by the intestinal macrophages are known to be rich in miRNAs [297,298], it may be of great interest to establish if they have anti-inflammatory and/or antioxidant functions [299] and develop innovative miRNA-based strategies to boost the impact of Solanaceae and cereals on human health.

## 8. Does a Diet including Antioxidant-Rich Cereals and *Solanaceae* Have an Impact on Human Health?

There is a growing interest in natural pigments as phytochemicals in food science and human nutrition, supported by preclinical, clinical, and large-scale population-based studies indicating their effect in preventing chronic diseases and suggesting a generalized concept according to which “the more coloured foods we eat the better the health benefits”. Cereals and *Solanaceae* have a dominant role in the human diet and improving their content of antioxidant compounds can have a strong impact on the total amount of antioxidant in human diet. A plethora of genetic studies have characterized the genetic diversity of these crops for content and composition in carotenoids and polyphenols and either traditional breeding or metabolic engineering works have been carried out in the last 20 years to select varieties with an increased content of beneficial compounds. Dedicated agronomic management practices and transformation processes can, to some extent, increase and preserve the antioxidant value of grains and fruits, but genetics (particularly genetic engineering) is the most robust strategy to enhance these compounds in edible parts of crops.

The data summarized in Table 1, Table 2 and Table 3 demonstrate the availability of cereals and *Solanaceae* varieties with an antioxidant content significantly higher than those of the most used varieties and preclinical studies support the potential benefits associated to the utilization of pigment-enriched varieties. Despite the strong evidence of pre-clinical and pilot studies (Table 4 and Table 5) suggesting that high polyphenol-enriched foods regimes lead to preventive effects against chronic diseases, an overall understanding of bioactivity, absorption/bioavailability and interactions among dietary bioactive compounds is still limited in terms of clinical studies. These considerations should influence that clinical studies are conducted to produce clear evidence on the potential impact of antioxidant-rich cereals- and *Solanaceae*-derived foods on human health.

Bioaccessibility and bioavailability are modified by many different factors, including food processing and mode of food consumption (cooked or fresh food, with or without fruit skin, etc.), chemical properties of the bioactive compounds, the ratios of the different phytochemicals in the ingested food matrix, other compounds present in the food matrix, and modifications which might occur during food digestion. To date, clinical studies (Table 5) are generally focused on specific single compounds and effects are studied on a limited number of markers; the study designs are different and are generally conducted on a small number of subjects with different characteristics (e.g., age, sex, nutritional status, ethnicity, etc.). Also, the influence of genetic factors on the response of individuals to specific phytochemicals must be considered. Further research is required to ascertain the impact of the consumption of antioxidant-rich cereals- and *Solanaceae*-derived foods on changes in microbiota and associated health benefits. Future studies should focus on understanding the complex interactions on bioaccessibility and bioactivity between multiple phytochemicals and other food components in different food matrices and further human intervention studies should be carried out with antioxidant-rich cereals- and *Solanaceae*-derived foods to strengthen the available evidence on their efficacy on preventing chronic diseases. Finally, to allow the comparison across the different studies, it is necessary to standardize the design of the experiments with specific reference to: (i) the adoption of common units for the definition of the phytochemicals amount; (ii) the type of extract and the diet administration in clinical studies; (iii) the clinical controls, both diet and model; and (iv) the amount scalability of antioxidant-rich food matrix from preclinical to clinical studies.

In conclusion, the application of pigmented cereals- and *Solanaceae*-derived foods in the reference healthy diet may lead to a new paradigm in public health nutrition and recommendation that is based on the use of long-term disease preventative measures. Nevertheless, an implementation of robust and comparable clinical trials on the effects of diets including antioxidant-rich cereals- and *Solanaceae*-derived foods is required.

The scientific gaps highlighted in this paper constitute the rationale of the FACCE JPI project: “SYSTEMIC—Knowledge hub on Nutrition and Food Security (systemic-hub.eu)”.

## Figures and Tables

**Figure 1 antioxidants-11-00794-f001:**
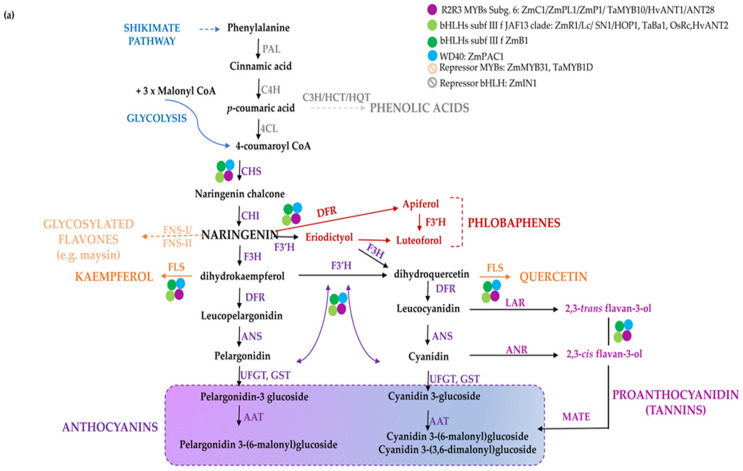
The flavonoid biosynthesis and its regulation in cereal and *Solanaceae* crops. Scheme of the pathway leading to the production of flavonoids and phenolic acids in monocots (**a**) and dicots (**b**). Phenylalanine is first deaminated by PAL to produce cinnamic acid, then converted by C4H into p-coumaric acid, which can enter the synthesis of hydroxycinnamic acids (i.e., chlorogenic acid and other phenolics) or it can be conjugated with coenzyme A to produce 4-coumaroyl-CoA by 4CL. CHS catalyses the condensation of p-coumaroyl-CoA with three molecules of malonyl-CoA to naringenin chalcone, then converted to the flavanone naringenin by CHI. Indeed, naringenin may be converted to flavones by FNSI/FNSII (e.g., maysin in maize), to the red phlobaphenes, derived from condensation of the 3-deoxy flavonoids apiferol and luteoforol (**a**) and to dihydroflavonols, such as dihydrokaempferol (DHK), which can then be used by F3′H to produce dihydroquercetin (DHQ) or by F3′5′H to form dihydromyricetin (DHM) (**b**). Dihydroflavonols are then converted to flavonols (e.g., kaempferol, quercetin, and myricetin) by FLS. Downstream, DFR reduces the dihydroflavonols to their respective colourless leucoanthocyanidins, which are then converted into the coloured anthocyanidins (e.g., cyanidin, pelargonidin, and delphinidin). The main enzymes catalyzing the reactions in the pathway are reported in violet. Regulatory proteins belonging to diverse classes of transcription factors are marked with coloured dots. Branches leading to different classes of flavonoids and anthocyanins are indicated with diverse colours; in B, thicker purple and red arrows highlight the branch leading to the most abundant derived anthocyanins. The name of the enzymes is detailed in the abbreviation list.

**Figure 2 antioxidants-11-00794-f002:**
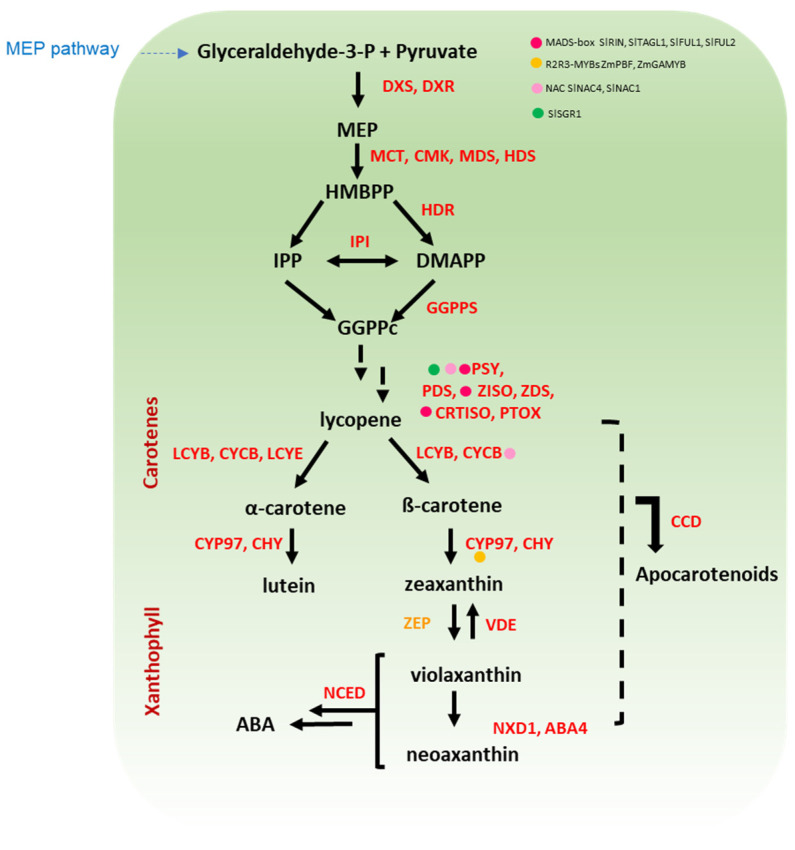
A simplified overview of carotenoid pathway in cereal and *Solanaceae* crops. The first step in carotenoid biosynthesis is the condensation of two GGPP molecules to form phytoene catalysed by *PSY*, which is the main rate-limiting step in solanaceous fruits and cereal grains. Further, the conversion of phytoene to lycopene via sequential desaturation and isomerization reactions is catalysed by a set of four enzymes (PDS, ZISO, ZDS, and CRTISO). Lycopene is at the branch point of carotenoid synthesis since it can be cyclized to ß-carotene or α-carotene by LCYB and LCYE. Downstream, the sequential hydroxylation and epoxidation of these carotenes leads to the production of diverse xanthophylls (e.g., lutein and zeaxanthin). Regulatory proteins belonging to diverse classes of transcription factors are marked with coloured dots. The name of the enzymes is detailed in the abbreviation list.

**Table 1 antioxidants-11-00794-t001:** Typical range of concentrations of phytochemicals in cereal crops. * no antioxidant effect since they are bound.

Antioxidant Class	Prevalent Compounds	Main Modifications	Distribution into the Grain	Heritability Range for Total Content	Rice (*O. sativa*)	Wheat (*T. aestivum*; *T. urum*)	Maize (*Z. mays*)	Sorghum (*S. bicolor* L.)
Carotenoids	Lutein, Zeaxanthin, β-carotene, β-cryptoxanthin		endosperm, aleurone and germ (cereals) [66]	0.7 in pigmented wheat and durum wheats [38] and in RILs of durum wheats [57]	negligible amount [67]	4–12 μg/g [49]	blue: 0.18 μg/g; yellow: 0.13–60 μg/g [68,69]	0.02 to 0.85 μg/g [70]
Vitamin E	α and β-tocopherol, α and β-tocotrienols		aleurone, pericarp and germ (maize)		19.36–63.29 μg/g [71]			2.81–29.62 μg/g [70]
Phenolic acids	ferulic acid, coumaric acid, syringic acid, vanillic acid, caffeic	mainly ester or ether linked to cell wall polimers	aleurone and germ (maize), bran, embryo and endosperm (rice)	0.63 in tetraploid wheat collection [39]	78.83–317.4 μg/g [72]	550–1700 μg/g; 27–45 μg/g [37,39]	blue: 2.60 mg/g; yellow: 3.2 mg/g; white: 2.6 mg/g [73]; purple: 2.66–17.5 mg/g [74]	1.0–29.6 mg/g [75]
Flavonoids	apigenin derivatives	mainly conjugated (glycosides)	pericarp (maize)		886–2863 (μg rutin equivalent/g) [76]	70–110 μg/g [36]	red: 27.53 mg/g [77]	* 0–23 mg/g [78]
Anthocyanins	cyanidin 3-O-glucoside	only conjugated (glycosides)	aleurone, pericarp, cob; pericarp (maize)	0.93 in pigmented wheat and durum wheats [38]	87.54 mg (Cyanidin-3-glucoside equivalent/100 g rice grain) [72]	purple: 8–50 μg/g; red: 1–25 μg/g; blue: 80–170 μg/g [38]	blue: 0.66 mg/g; purple: 1.64 mg/g [79]; purple cob: 3.1–12.6 mg/g [79,80]	1–3 mg/g [35]

**Table 3 antioxidants-11-00794-t003:** Typical range of concentration of phytochemicals in naturally biofortified and metabolic engineered cereal and *Solanaceae* Crops.

	Concentration of Phytochemicals (mg/g) Cereal and *Solanaceae* Crops Obtained by Breeding or Metabolic Engineering			
Antioxidant Class	Rice (*O. sativa*)	Tomato (*S. lycopersicum*)	Potato (*S. tuberosum*)	Eggplant (*S. melongena*)
Carotenoids	Golden rice: 5.06 μg/g, Golden Rice 2: up to 37 µg/g DW [89]	SRG1 mutants 5.1× lycopene [126]; “Sun Black” (*Aft*/*Aft atv*/*atv*, peel) 0.2 mg/g DW total carotenoid content [127]; “Bronze” (*E8*:*MYB12*, *E8*:*Del*/*Ros*, *35S*:*StSy*) ∼0.55 mg/g DW total carotenoid content [132]	Golden potato (cv. Desiree) 3600-fold increase in beta carotene to 4.7 mg/100 g DW [146]; 4 mg/100 g DW zeaxanthin in *S. tuberosum* 4n [147]. From traces to 0.33 mg/100 g beta-carotene [148]	eggplant transgenic line *EEF48*:*crtB* 0.15 mg g^−1^ FW of β-carotene [149]
Phenolic acids		“Sun Black” (*Aft*/*Aft atv*/*atv*, peel) 0.6 mg/g DW of CGA, 8.6 mg/g DW total phenolic content [127]; “Yellow” *E8*:*MYB12* 15 mg g^−1^ DW CQAs equivalent to 22-fold higher levels, respectively [150]	3.35-fold increases on average) [151]	
Flavonoids (considered as total content)		“Sun Black” (*Aft*/*Aft atv*/*atv*, peel) 0.8 mg/g DW rutin [132], “Yellow” (*E8*:*MYB12*) 72 mg g^−1^ DW total flavonols equivalent to 65-fold higher levels [150]; “Bronze” (*E8*:*MYB12*, *E8*:*Del*/*Ros1*, *35S*:*StSy*), “Indigo” (*E8*:*MYB12*, *E8*:*Del*/*Ros1*) ∼15 and 20 mg/g DW total flavonols, respectively [132]	Flavonols (4.50-fold increase on average) [151]	
Flavonoids (Anthocyanins)		“Sun Black” (*Aft*/*Aft atv*/*atv*) 1.2 mg/g DW in fruit peel [127]; *Del/Ros1* (*E8*:*Del*/*Ros1*) 14.7 mg g^−1^; “Indigo” (*E8*:*MYB12*, *E8*:*Del*/*Ros1*) ∼5–24 mg g^−1^ DW [131,132]; “Crimson” (*E8*:*Del*/*Ros1*, *E8*:*AmDFR*, *f3′5′h*) 5.3 ± 1.3 mg/g DW, “Magenta” (*E8*:*Del*/*Ros1*, *E8*:*MYB12, E8*:*AmDFR*, *f3′5′h*) 7.9 ± 2.3 mg/g DW total anthoyanins [131]	From 0.4 in wt to 3 ug/100 mg Petunidin (7×); from 0.04 in wt to up to 0.3 ug/100 g Pelargonidin (7×) [152]	

**Table 4 antioxidants-11-00794-t004:** Health effects of antioxidant-rich foods in preclinical studies. HFD, high-fat diet; HCD, high cholesterol diet; ATM, adipose tissue macrophages; Alzheimer’s disease (AD); Parkinson’s disease (PD); CCl4, carbon tetrachloride; MNU1, methyl-1-nitrosourea; GalN, D-galactosamine; and STZ, streptozotocine.

	Antioxidant-Rich Food	Health Effects in Animal Models	Refs.
*Cereals*	Black rice	Improved hyperlipidemia and insulin resistance in rats on high-fructose diet	[219]
		Reduced hyperlipidemia in rats fed HCD	[220]
		Reduced dyslipidemia, induced optimal platelet function in rats fed HFD	[221]
		Regulated cholesterol metabolism and improved dysbiosis of gut microbiota in mice fed HCD	[222]
		Reduced ethanol-induced liver damage in rats	[227]
		Attenuated liver injury and prevented fibrosis in CCl4-treated mice	[228]
		Ameliorative effect in senescent mice induced by D-galactose	[229]
	Black/red rice	Decreased atherosclerotic plaques, increased antioxidant status in rabbit fed HCD and in apoE-deficient mice	[223,224,225]
	Purple rice	Prevented neurodegeneration in a rat model of AD	[230]
	Black/purple wheat	Prevented obesity, hyperlipidemia, and insulin resistance in mice fed HFD	[226]
	Purple wheat	Preventive effect on cognitive functions in mouse models of AD and PD	[231]
	Blue corn	Reduced cardiac infarct size following ischemia-reperfusion in rats	[233]
	Purple corn	Long lasting cardioprotection against ischemia-reperfusion mediated by microbiota in mice	[234]
		Increased survival and reduced cardiac damages against Doxorubicin-induced cardiotoxic effects in mice	[235]
		Reduced diabetes-associated renal fibrosis, angiogenesis, and mesangial and glomerulal inflammation in *db/db* mice	[236,237]
		Prevented obesity and ameliorated hyperglycemia in mice fed HFD	[217,218,219]
		Reduced obesity-associated inflammation by reprogramming of ATM in mice fed HFD	[77]
		Reduced trigeminal inflammatory pain in rats	[239]
		Delayed progression of muscular dystrophy reducing inflammation and oxidative stress in Sgca null mice	[240]
*Solanaceae*	Purple tomato	Delayed cancer development and increased life span in *p53^-/-^* mice	[120]
		Reduced inflammation and induced antioxidant response in rat model of carrageenan-induced paw oedema	[244]
	Bronze tomato	Reduced inflammation markers, modulated gut microbiota in Winnie mice	[132,246]
	Purple potato	Prevented gastrointestinal inflammation/cancers in pig fed HFD	[247]
		Reduced chronic intestinal inflammation in DSS-induced colitis in mice	[248,249]
		Prevented obesity, hyperlipidemia, and insulin resistance in rats fed HFD	[252,253]
		Attenuated hyperglycemia in STZ-induced diabetic rats.	[254]
		Reduced obesity-associated oxidative damage in rats fed HCD	[256]
		Suppressed GalN-induced hepatotoxicity via inhibition of lipid peroxidation and/or inflammation in rats	[255]
	Purple/red potato	Reduced proliferation of the benzopyrene-induced stomach cancer in mice	[250]
	Red potato	Reduced MNU1-induced breast carcinogenesis in rats	[251]
		Inhibited hepatic lipid peroxidation in rats	[257]

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
