# Peer review of "Does Plant Breeding for Antioxidant-Rich Foods Have an Impact on Human Health?"

_antioxidants, 2022, doi:10.3390/antiox11040794_

Round 1

Reviewer 1 Report

The review is interesting as it describes in detail the polyphenol content of several foods commonly consumed all around the world, providing also information about possible polyphenol-enriched novel food  that could be  useful preventive strategies against a number of diseases. The authors also mention some critical point such as the bioavailability of these compounds, the effects of cooking and processing, and the relationship with microbiota.

Few points should be improved

  • In the introduction  the well-known activity of polyphenols as regulator of gene expression and enhancer of endogenous antioxidant defense system should be mentioned. The lines 567-583, thus, should be moved to introduction.
  • The section 2 should be shortened; the figures provide already enough details on this issue.
  • One or two (1 for preclinical and1 for clinical studies) tables  should be organized to summarize the studies reported in section 6; this would facilitate the readers.  

Reviewer 2 Report

There are a number of corrections that need to be made.  Please see the attached document.
